# Characterisation of an unusual nicotinic acetylcholine receptor subtype preferentially sensitive to biogenic amines
Eleanor L. Mitchell [1,5], Emily B. Armstrong [2,5], Franco Viscarra [2,3], Isabel Bermudez[2], Philip C. Biggin [3], James A. Goodchild [4] & Andrew K. Jones [2] ✉

Nicotinic acetylcholine receptors (nAChRs) are best known for mediating the fast actions of acetylcholine. However, the spectrum of other neurotransmitters possibly acting on these receptors is not well understood. Here, we report that the α5 nAChR subunit of the honey bee, *Apis mellifera*, when expressed in *Xenopus laevis* oocytes, has unusual pharmacological properties in that it has high sensitivity towards dopamine, tyramine and octopamine with $EC_{50}$ values of 3.37 μM, 91.1 μM and 378 μM, respectively, whereas the $EC_{50}$ for acetylcholine is 2.37 mM. The biogenic amines are also considerably more efficacious than acetylcholine in activating the receptor. Molecular dynamics simulations and expression of α5 mutants identify the lack of a proline doublet in loop E as playing a major role in determining dopamine efficacy. Together with phylogenetic analysis using homologous receptors from other species, this study enhances our understanding of how ligand-gated ion channels evolve functional diversity.

Nicotinic acetylcholine receptors (nAChRs) mediate the fast actions of acetylcholine (ACh) in the nervous system and are members of the cys-loop ligand-gated ion channel (CysLGIC) superfamily, which also includes receptors for γ-aminobutyric acid (GABA), glycine, serotonin (5-HT₃ receptors), glutamate and histamine[1,2]. As with other CysLGICS, nAChRs consist of five subunits arranged around a central ion channel, which is opened upon binding of ACh[3]. Each nAChR subunit is encoded for by a separate gene and possesses an N-terminal extracellular region as well as four transmembrane domains. The ligand-binding site is formed by six distinct regions (loops A–F) located in the N-terminal region[4]. Subunits possessing two adjacent cysteine residues in loop C, which are important for ACh binding[5], are denoted as α type whilst subunits lacking these two cysteines are referred to as non-α[4]. A minimum of two α subunits are required to form a functional nAChR and receptors can either be homomeric, consisting of the same α subunit, or heteromeric, which are made up of two or more different subunits.

Insect nAChRs are of interest as they are targets of highly effective insecticides[6]. Analysis of genome sequences from a variety of species have identified complete nAChR gene families[7–9], commonly consisting of 10–12 subunit genes although examples of larger gene families have been found in the parasitoid wasp, *Nasonia vitripennis* (16 subunits)[10] and the cockroaches *Blattella germanica* (17 subunits) and *Periplaneta americana* (19 subunits)[11]. Core groups of nAChR subunits that are highly conserved in insects have been identified, where α1, α2, α3, α4, α5, α6, α7, α8 (or β2 in some Dipteran species) and β1 subunits show clear orthologous relationships between different species[7]. Intriguingly, phylogenetic analysis shows that the α5 subunit of Dipteran species (such as *Aedes aegypti*[12], *Anopheles gambiae*[13], *Drosophila melanogaster*[7] and *Musca domestica*[14]) forms a subgroup that is distinct from α5 subunits of non-Dipteran species including Blattodea (*B. germanica* and *P. americana*[11]), Coleoptera (*Tribolium castaneum*[15]), Hymenoptera (*A. mellifera*[16], *Bombus terrestris*[17] and *N. vitripennis*[10]) and Lepidoptera (*Bombyx mori*[8] and *Cydia pomonella*[18]). Whereas Dipteran α5 subunits are closely related to α7 subunits, the non-Dipteran α5 subunits do not appear to have a clear association with any other subunit despite having sequence hallmarks characteristic of α subunits, namely the two adjacent cysteine residues in loop C[19]. When heterologously expressed in *Xenopus laevis* oocytes, the α5 subunit of *D. melanogaster* formed a functional homomeric receptor with an $EC_{50}$ for ACh at 8.8 μM, indicating a sensitivity towards ACh that is typical of nAChRs[20]. Similarly, heteromeric nAChRs consisting of various combinations of α1, α2, α8 or β2 and β1 subunits from *D. melanogaster*, *A. mellifera* or *B. terrestris* showed sensitivities

[1]School of Biomedical Sciences, Faculty of Biological Sciences, University of Leeds, Leeds, UK. [2]Department of Biological and Medical Sciences, Faculty of Health and Life Sciences, Oxford Brookes University, Headington, Oxford, United Kingdom. [3]Structural Bioinformatics and Computational Biochemistry, Department of Biochemistry, University of Oxford, Oxford, United Kingdom. [4]Syngenta, Jealott's Hill International Research Centre, Berkshire, United Kingdom. [5]These authors contributed equally: Eleanor L. Mitchell, Emily B. Armstrong. ✉e-mail: a.jones@brookes.ac.uk

typical of nAChRs with $EC_{50}$ values for ACh in the lower micromolar range[21,22]. In contrast, the α5 subunit of *A. mellifera* was considerably less sensitive to ACh with an $EC_{50}$ of 2.37 mM suggesting that other neurotransmitters may act as endogenous ligands[19]. Co-expressing α5 with other *A. mellifera* nAChR subunits did not affect sensitivity to ACh, indicating it may not be contributing towards the function of heteromeric receptors. Serotonin was found to be a more potent agonist with an $EC_{50}$ of 119.0 μM that was also more efficacious than ACh in activating the receptor demonstrating that honey bee α5 possesses unusual pharmacological properties for a nAChR. Here, we report further functional and pharmacological characterisation of this receptor in order to explore its evolutionary relationships with homologous receptors in other species. We show that the biogenic amines, dopamine, tyramine and octopamine, act as superagonists relative to ACh on the homomeric *A. mellifera* α5 receptor revealing the non-Dipteran α5 subunits as being a potentially distinct subclass of CysLGICs present in certain orders of insects. Using in silico simulations and the use of α5 mutants, we show that loop E plays a major role in determining biogenic amine actions.

## Results

### Dopamine, octopamine and tyramine act as superagonists on *A. mellifera* α5

With serotonin showing more potent agonist activity than ACh on the *A. mellifera* α5 nAChR[19], we measured the actions of other biogenic amines on the honey bee receptor expressed in *X. laevis* oocytes. We found that dopamine acted as an agonist with current amplitude being concentration dependent whilst no responses were detected in oocytes injected with water only (Fig. 1a). An $EC_{50}$ of 3.37 (2.926–3.885) μM was determined from a concentration response curve, which is significantly smaller than that of ACh with 2653 (2405–2926) μM (Fig. 1c and Table 1). The mean relative maximum current response at 100 μM dopamine was 3530% of the mean relative maximum current response to 5 mM ACh (Fig. 1b and Table 1). Likewise, octopamine acted as an agonist with an $EC_{50}$ of 378 (354.3–403.5) μM (Fig. 1c) and maximum current responses (at 1 mM) being 899% of the current responses to 5 mM ACh (Fig. 1b and Table 1). Tyramine also elicited current responses in a concentration-dependent manner with an $EC_{50}$ of 91.1 (78.31–106.0) μM (Fig. 1c). It, too, showed superagonist actions with maximum current responses (at 1 mM) being 770% of the current responses to 5 mM ACh (Fig. 1b and Table 1).

### Histamine acts as a partial agonist on *A. mellifera* α5

Histamine showed agonist actions on the *A. mellifera* α5 nAChR expressed in *X. laevis* oocytes, eliciting responses in a concentration-dependent manner. However unlike the other biogenic amines tested, the receptor did not show significant difference in sensitivity to histamine compared to ACh as shown by the $EC_{50}$ value of 3355 (2278–4941) μM (Fig. 1c and Table 1). Histamine also acted as a partial agonist since it evoked maximum current responses (at 5 mM) of 45.3% of the maximum ACh response (5 mM) (Fig. 1b and Table 1).

### *A. mellifera* α5 did not respond to GABA, glutamate, glycine or homovanillyl alcohol

We tested to see whether GABA, glutamate or glycine were able to activate the *A. mellifera* α5 nAChR expressed in *X. laevis* oocytes. We found that at 1 mM none of them were able to induce detectable responses (Fig. 2a–c). We also tested homovanillyl alcohol, which has a structure similar to dopamine and is a component of the honey bee queen mandibular pheromone[23] and found that concentrations at 100 μM or 1 mM did not elicit a response (Fig. 2d).

### *A. mellifera* α5 was unresponsive to amitraz but was antagonised by α-bungarotoxin

With the *A. mellifera* α5 nAChR responding to octopamine (Fig. 1), we tested to see whether the insecticide and arachnicide[24], amitraz, activates the receptor since it has been classified as an agonist of octopamine receptors[25].

We found that amitraz did not show any agonist actions on the *A. mellifera* α5 nAChR expressed in *X. laevis* oocytes nor antagonistic actions when coapplied with 400 μM octopamine (Fig. 2e).

Since honey bee nAChRs have been commonly divided into α-bungarotoxin sensitive or insensitive receptors[26,27], we tested to see whether it acted on the *A. mellifera* α5 nAChR. We found that 0.1 μM α-bungarotoxin showed antagonistic actions on the *A. mellifera* α5 nAChR expressed in *X. laevis* oocytes. The ACh response was reduced by pre-incubation with 0.1 μM α-bungarotoxin and by co-application at 2 mM with 0.1 μM α-bungarotoxin. The response to 2 mM ACh was restored after 26 min washing (Fig. 2f).

### *A. mellifera* α5 is a sodium-permeable ion channel

We tested the I–V relationship across a voltage range of −100 to +40 mV in oocytes expressing the *A. mellifera* α5 nAChR and found that conductance progressively depolarises from −80 mV (Fig. 3) as is typical for a nAChR[28]. The ion selectivity of the *A. mellifera* α5 nAChR was tested by substituting the NaCl in SOS with N-methyl-D-glucamine chloride (NMDG) and measuring I–V relationship. The absence of $Na^+$ eliminated the response to $EC_{50}$ concentrations of ACh or dopamine, indicating that sodium ions participate in the agonist-induced response and thus reinforcing the *A. mellifera* α5 nAChR as being cation selective (Fig. 3).

### Co-application with dopamine modulated the response of *A. mellifera* α5 to acetylcholine

Application of 1 μM dopamine or 500 μM ACh to oocytes expressing *A. mellifera* α5 did not induce detectable responses (Fig. 4a). However co-application of both agonists at these concentrations resulted in a notable response, indicating synergistic agonism. We found that varying concentrations of ACh with 1 μM dopamine generated a biphasic concentration response curve, with concentrations from 3 μM to 5 mM of ACh resulting in increasing response and higher doses of 7 and 10 mM in inhibited responses (Fig. 4b) in line with increased agonist concentrations leading to a desensitised state.

### Molecular dynamics simulations of *A. mellifera* α5

In order to identify structural components of the *A. mellifera* α5 nAChR important for biogenic amine sensitivity, a three-dimensional model of the honey bee receptor was generated using the human α7 nAChR structure as a template (Supplementary Fig. 1). Representative structures of ACh and dopamine bound to the binding site extracted by clustering show similar conformations for both ligands, with the charged amino group pointing towards the highly conserved tryptophan (W172) of loop B and the rest of the molecule oriented towards the complementary chain (Fig. 5a, b). In terms of the loop C conformation, dopamine stabilised the loop at a distance just less than 14 Å, on the other hand, ACh stabilised the loop C distance to a bimodal distribution with two peaks, one at ~14 Å and another, at ~17 Å (Fig. 5c). The structures obtained from clustering were used as a starting point for binding free energy (BFE) estimations, which resulted in a BFE of −8 ± 0.40 kcal/mol for ACh and −10 ± 0.77 kcal/mol for dopamine that are significantly different (Fig. 5d), thus supporting the preferences of *A. mellifera* α5 nAChR for dopamine. *A. mellifera* α5 shares 38% sequence identity with the human α7 nAChR, which relates to a low conservation in the complementary chain of the binding site (Supplementary Fig. 2). Loops C and E share 56% and 45% identity, respectively. Notably, in the human α7, a conserved pair of proline residues disrupts the secondary structure of loop E. In contrast, in the *A. mellifera* α5 there is a serine and histidine in their place, which can effectively engage in backbone hydrogen bond interaction with neighbouring residues, stabilising the sheet structure. To test how this difference impacts the binding mode of the ligands, simulations were carried out with the S143P + H144P *A. mellifera* α5 double mutant. Additionally, the H144P substitution was also simulated in line with functional results (see below and Table 2). In both cases, the distribution of loop C distances in the presence of dopamine was shifted towards a more open state of ~15 Å for the single mutant and a bimodal distribution with modes at ~15 and ~19 Å

**Fig. 1 | Responses to biogenic amines in *X. laevis* oocytes expressing the *A. mellifera* α5 nAChR subunit. a** Representative current traces showing responses to different concentrations of dopamine (DA) (300 nM–300 µM). The first trace shows the response of an oocyte injected with water only whilst the remaining traces show responses from oocytes injected with α5. **b** Sample traces comparing the response to maximal concentrations of ACh (5 mM) and dopamine (DA), octopamine (OA), tyramine (TA) or histamine (HA) from the same oocyte. **c** Concentration response curves of acetylcholine (ACh), dopamine (DA), octopamine (OA), tyramine (TA) and histamine (HA). Data are normalised to concentrations shown in Fig. 1b and are from 5 to 6 different eggs from 3 to 4 batches of frogs. Error bars represent SEM. Scale bars represent time in seconds (X axis) and current in nA and µA (Y axis).

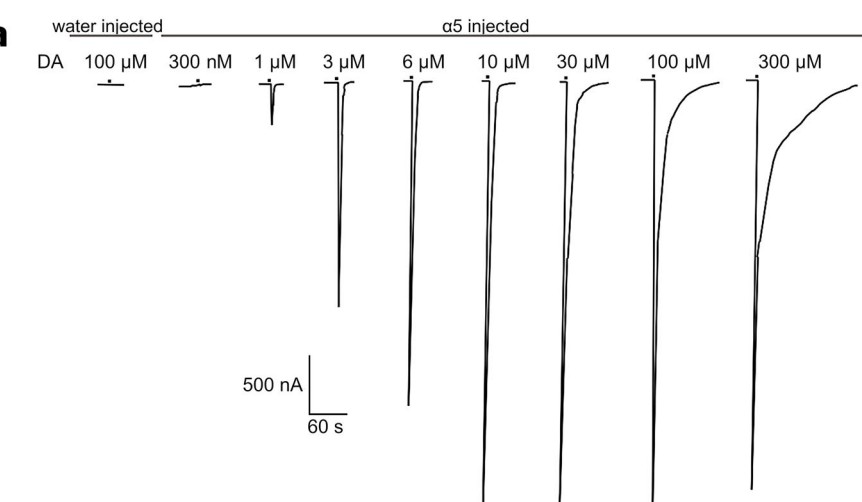

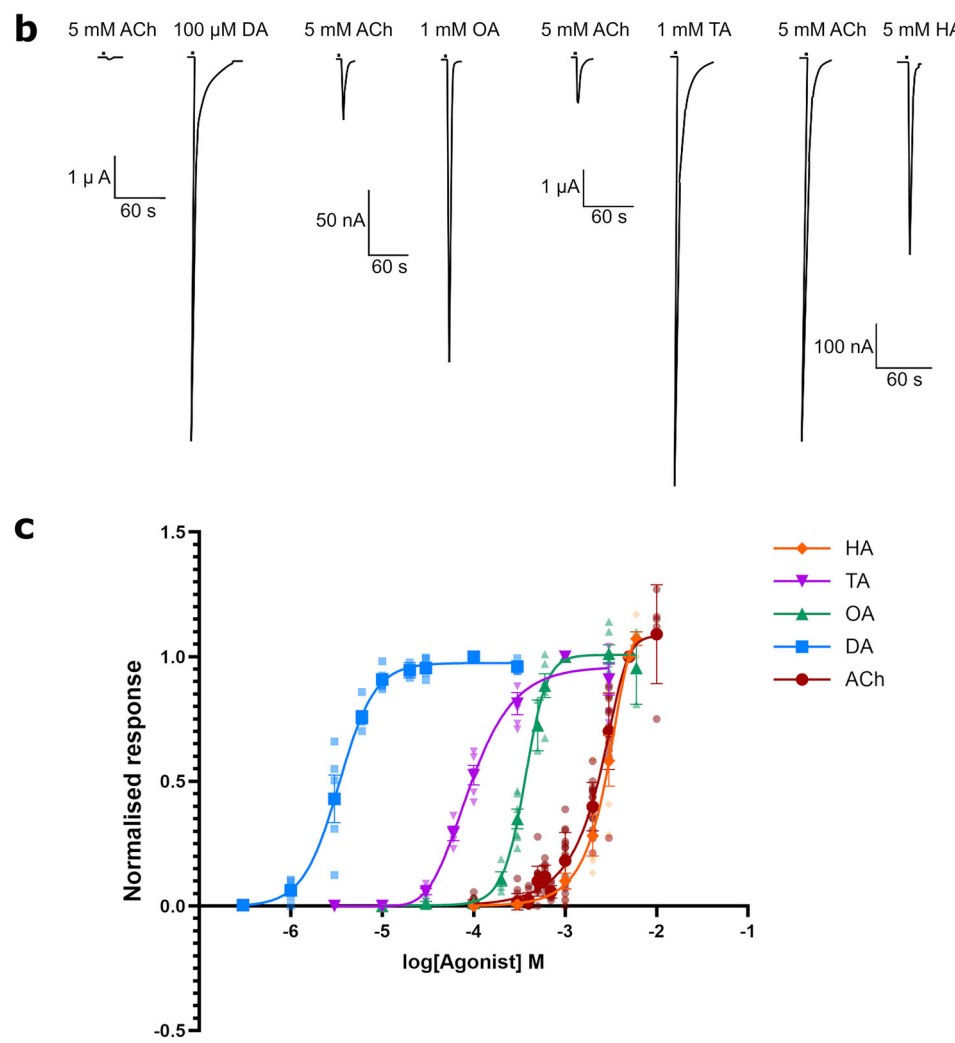

(Fig. 5c). On the other hand, the distance distributions for the mutants with ACh remained mostly unaffected. This can also be seen with ACh simulations showing high overlapping distance distributions (>%60) between wild-type and mutant α5 nAChRs compared to those with dopamine (~30%) (Supplementary Fig. 3). Furthermore, BFE calculations for the mutant honey bee nAChR revealed that substituting in the proline doublet in loop E inverted the preference of the receptor for dopamine (Fig. 5d).

## Loop E is important for determining the activity of dopamine

In order to verify predictions made through analysis of *A. mellifera* α5 homology models, site-directed mutagenesis was performed to generate mutants of the receptor with single amino acids in loop E substituted by the equivalent residues found in the human α7 nAChR (Supplementary Fig. 2). Also, overlapping PCR was used to generate honey bee α5 subunits with either the whole of loop E or loop C, or both, being substituted by the

**Table 1 | Responses of the *A. mellifera* α5 nAChR expressed in *X. laevis* oocytes to different neurotransmitters arranged according to EC$_{50}$ values**

| Agonist | EC$_{50}$ µM | Hill coefficient | Maximum response relative to 5 mM ACh (%) |
|---|---|---|---|
| Dopamine | 3.371 (2.926–3.885) ***$p = 0.0008$ | 2.220 (1.585–2.855) | 3530 (1630–5430) ****$p = 0.0001$ |
| Tyramine | 91.12 (78.31–106.0) $p = 0.0820$ | 1.918 (1.355–2.481) | 770 (633–907) *$p = 0.0369$ |
| Serotonin[a] | 121.6 (110.7–133.5) *$p = 0.0109$ | 3.419 (2.503–4.335) | 238 (214–262) $p > 0.9999$ |
| Octopamine | 378.1 (354.3–403.5) $p = 0.0525$ | 3.727 (2.998–4.455) | 899 (509–1289) *$p = 0.0326$ |
| Acetylcholine[a] | 2653 (2405–2926) | 2.304 (1.886–2.721) | 100 |
| Histamine | 3355 (2278–4941) $p > 0.9999$ | 2.500 (1.050–3.950) | 45.3 (34.5–56.1) $p > 0.9999$ |
| Choline[a] | 9070 (4482–18350) $p > 0.9999$ | 2.043 (0.9065–3.180) | 100 $p > 0.9999$ |

EC$_{50}$ values and Hill coefficients are displayed as the mean (95% confidence limits) and maximum responses displayed as percentages ± 95% confidence limits. All data are from 5 to 18 different oocytes from ≥3 batches of eggs. Statistical test used was one-way ANOVA with Bonferroni's multiple comparisons test.
*Indicates a significantly different value to that of acetylcholine ($p \leq 0.05$), **($p \leq 0.01$), ***($p \leq 0.001$) and ****($p \leq 0.0001$).
[a]EC$_{50}$ values for these agonists were initially reported previously[19], but determined according to the 'Methods' here to enable comparison of all data.

equivalent loops from the human α7 subunit. The resulting *A. mellifera* α5 mutants were expressed in *Xenopus* oocytes to measure responses to ACh and dopamine (Table 2). ACh or dopamine did not elicit detectable responses in several mutants that included S143P (Table 2), suggesting that this amino acid change may have rendered the receptor non-functional. Substituting the whole of loop E, which includes S143P, also resulted in a non-responsive receptor. Responses, however were observed in the *A. mellifera* α5 mutant with human loop C, which was activated by ACh or dopamine (Fig. 6). Whilst the EC$_{50}$ values for ACh and dopamine were not significantly altered (Table 2), the efficacy of dopamine was significantly reduced, from being 14.94 times greater than the response to ACh down to 2.376 (Table 2 and Fig. 6). Responses were also detected for the mutants E138H and V140Q, where there was no significant decrease in EC$_{50}$ of ACh and the size of response elicited by 100 µM dopamine were still considerably larger than those elicited by 5 mM ACh (Table 2). For E138H, there was a significant increase in ACh and dopamine EC$_{50}$s as shown by rightward shifts in Fig. 6a, b. In contrast, the H144P mutant showed greater sensitivity to ACh than wild-type as shown by a significantly smaller EC$_{50}$ of 434.7 µM (Table 2 and Fig. 6a) whilst with dopamine responses were too small to construct a concentration response curve (Fig. 6c). Also, H144P caused dopamine to switch from being a superagonist to now showing partial agonist actions as shown by a smaller response to dopamine than to ACh (Table 2 and Fig. 6c).

## Discussion

We report here that the *A. mellifera* α5 nAChR has unusual pharmacology in that the biogenic amines dopamine, octopamine and tyramine act as superagonists, evoking responses considerably greater than to ACh, with also greater potency as shown by lower EC$_{50}$ values (Table 1). Dopamine was the most efficacious and potent, eliciting a response up to 35 times greater than that of ACh and with an EC$_{50}$ of 3.37 µM (Fig. 1). The finding that dopamine, tyramine, octopamine and serotonin have EC$_{50}$ values in the micromolar range (Table 1) indicates that biogenic amines may act as endogenous neurotransmitters on *A. mellifera* α5.

Molecular dynamics simulations showed that in the case of dopamine (Fig. 5a), three cation-π interactions were observed, two with tyrosine residues (213 and 220) located in loop C and one with tryptophan 172 in loop B. Additionally, there is a hydrogen bond between the ligand amine and the main chain of tryptophan 172, a π-π interaction between the aromatic ring of dopamine and tryptophan 77, a hydrogen bond between one of dopamine's hydroxyl groups and hydrophobic interactions with leucine 142 and cysteine 215. On the other hand, ACh (Fig. 5b) has three cation-π interactions with tyrosines 213 and 220 and tryptophan 172 and hydrophobic interactions with tryptophan 77 and leucine 142. More interactions

of the receptor with dopamine compared to ACh may explain the difference in the free energy calculated by the QM/MM GBSA method (Fig. 5d), while this relates with the difference in affinities, it does not account for the difference in efficacies (Table 1). What could explain this difference is the stabilisation of different conformations of the protein, such as the so called 'capping' of the binding site by loop C, which has been associated with agonist bound structures in nAChRs[29,30]. Furthermore, this motion is part of the transition of the receptor to the high affinity state, a necessary step for ligand induced activation[31]. Therefore, less effective stabilisation of the capped loop C conformation can be related to less activation, such as for ACh compared to dopamine (Table 1). Considering that the amino acids of the (+) face of the binding site are mostly conserved and that the (−) subunit has been observed to be important for agonist selectivity in the α7 nAChR[30], the variation in the amino acids of the complementary face of the binding site are suspected to be the determinants of the unique pharmacology of the *A. mellifera* α5 receptor (Supplementary Fig. 2). One striking difference is Ser143 and His144 in loop E, which correspond to Pro142 and Pro143 in the vertebrate α7 receptor, a proline doublet that is highly conserved in nAChRs from both vertebrates and invertebrates. Our functional data with the *A. mellifera* α5 H144P mutant (Table 2 and Fig. 6) shows that this difference plays a considerable part in determining dopamine efficacy and possibly potency. Since proline cannot be a hydrogen bond donor, it prevents the formation of secondary structures[32], a β-sheet in this case, which confers greater flexibility to the loop. Analysis of selected distances to residue 144 in the wild-type (histidine) and mutant (proline) proteins shows that it interacts less tightly with loop D, present in the adjacent subunit right across loop E, while staying closer to loops A and B (Supplementary Fig. 4). Another important difference can be observed in the top portion of loop E, where His137 and Gln139 in α7 are replaced by Glu138 and Val140, respectively, in the honey bee α5 receptor (Supplementary Fig. 2). This change of a polar residue to a hydrophobic one and a positively charged amino acid to a negatively charged one further contributes to a preference for aromatic side groups in ligands through alkyl-π and anion-π interactions. Another difference, where Val139 in honey bee α5 is Cys138 in vertebrate α7, was not tested individually based on the observation that V139 has its side chain orientated away from the binding site, so we did not expect a change in this residue would alter the interactions of the binding site with the ligand. Finally some amino acid differences in loop C of the *A. mellifera* α5 subunit could affect the dynamics that is essential for receptor gating[33,34]: Glu211, which replaces Arg208 in the α7 receptor, Pro217 that replaces Lys214 and Ser214 replaces Glu211.

Biogenic amines have been found to elicit responses from other members of the CysLGIC superfamily in invertebrates. For instance, dopamine, octopamine, serotonin and tyramine at 1 mM were found to

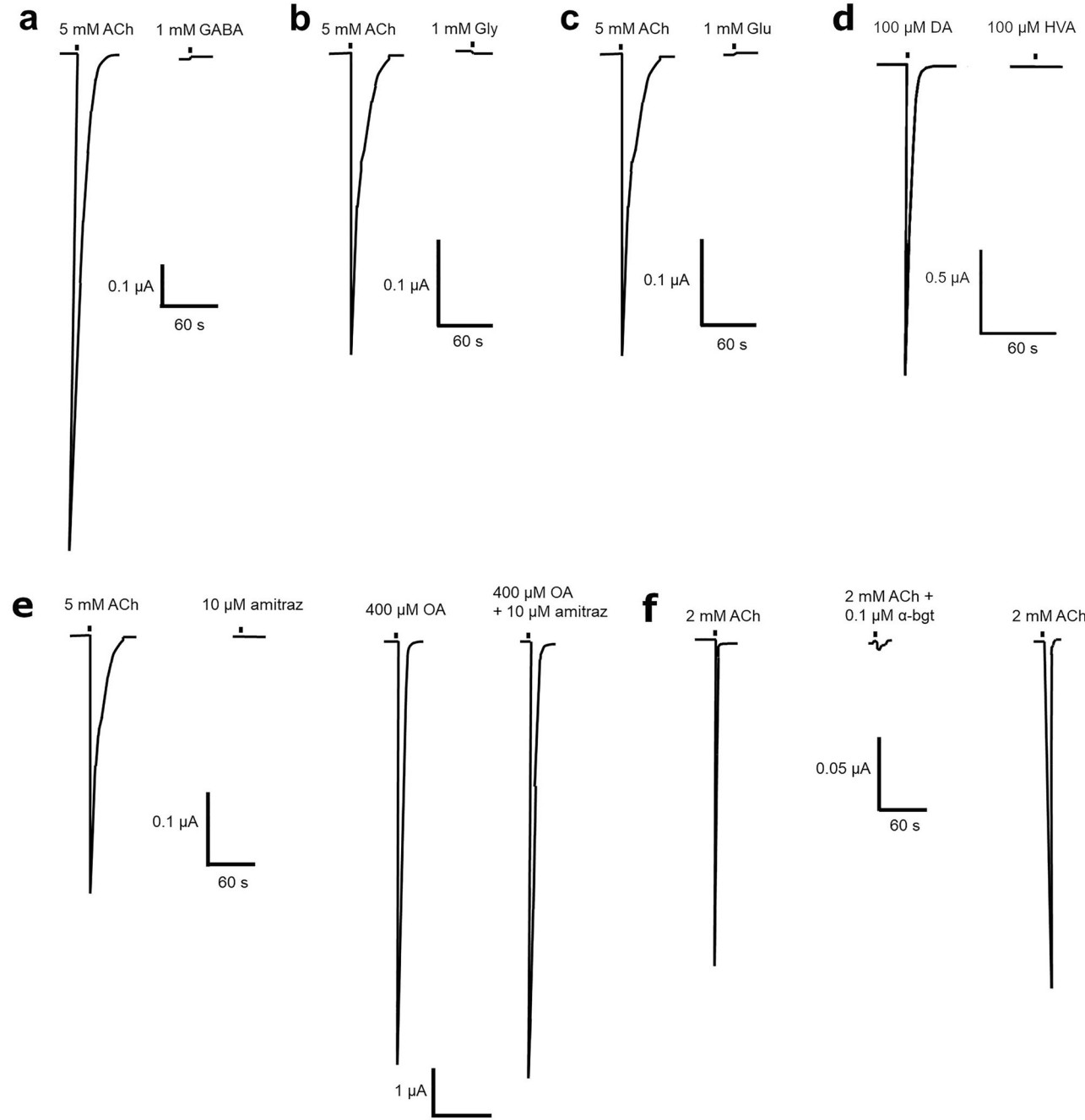

**Fig. 2 | Responses to GABA, glycine, glutamate, homovanillyl alcohol, amitraz and α-bungarotoxin in *X. laevis* oocytes expressing the *A. mellifera* α5 nAChR subunit.** Representative current traces showing that **a** 1 mM GABA, **b** 1 mM glycine (Gly) or **c** 1 mM glutamate (Glu) has no agonist actions when applied to oocytes expressing the *A. mellifera* α5 nAChR 3 min after 5 mM acetylcholine (ACh) application. **d** Representative current traces showing that homovanillyl alcohol (HVA) has no agonist actions when 100 μM was applied to oocytes expressing the *A. mellifera* α5 nAChR 3 min after 100 μM dopamine (DA) application.

**e** Representative current traces showing that amitraz has no agonist actions when 10 μM was applied to oocytes expressing the *A. mellifera* α5 nAChR 3 min after 5 mM acetylcholine (ACh) application and that amitraz has no antagonist actions when 10 μM was coapplied with 400 μM octopamine (OA). **f** Representative current traces showing responses from oocytes to 2 mM ACh, 2 mM ACh coapplied with 0.1 μM α-bungarotoxin (α-bgt) after 3 min 20 s incubation with 0.1 μM α-bgt, normal response to 2 mM ACh restored after 26 min washing with SOS. Scale bars represent time in seconds (X axis) and current in μA (Y axis).

activate the histamine-gated chloride channel (HisCl1 or HCLB) of the house fly, *Musca domestica*, although with current sizes considerably smaller than that of histamine[35]. Dopamine and tyramine were found to be potent agonists of several anionic CysLGIC subunits (LGC-51, LGC-52, LGC-53, LGC-54, LGC-55 and LGC-56)[36] from the nematode, *Caenorhabditis elegans*, with EC$_{50}$s in the micromolar range[37,38]. Phylogenetic analysis shows that the *C. elegans* subunits form a distinct group that does

not include the *A. mellifera* α5 subunit (Fig. 7) indicating that, despite being gated by dopamine and tyramine, they are unrelated and are likely the result of a different evolutionary trajectory having originated from anion-permeable cysLGICs[36]. Instead, *A. mellifera* α5 clusters with α5 nAChR subunits from other non-Dipteran insects[19] including Blattodea (*P. americana*), Coleoptera (*T. castaneum*), Lepidoptera (*B. mori*) and Orthoptera (*Locusta migratoria*) (Fig. 7). It has been speculated that the α5 subunit in

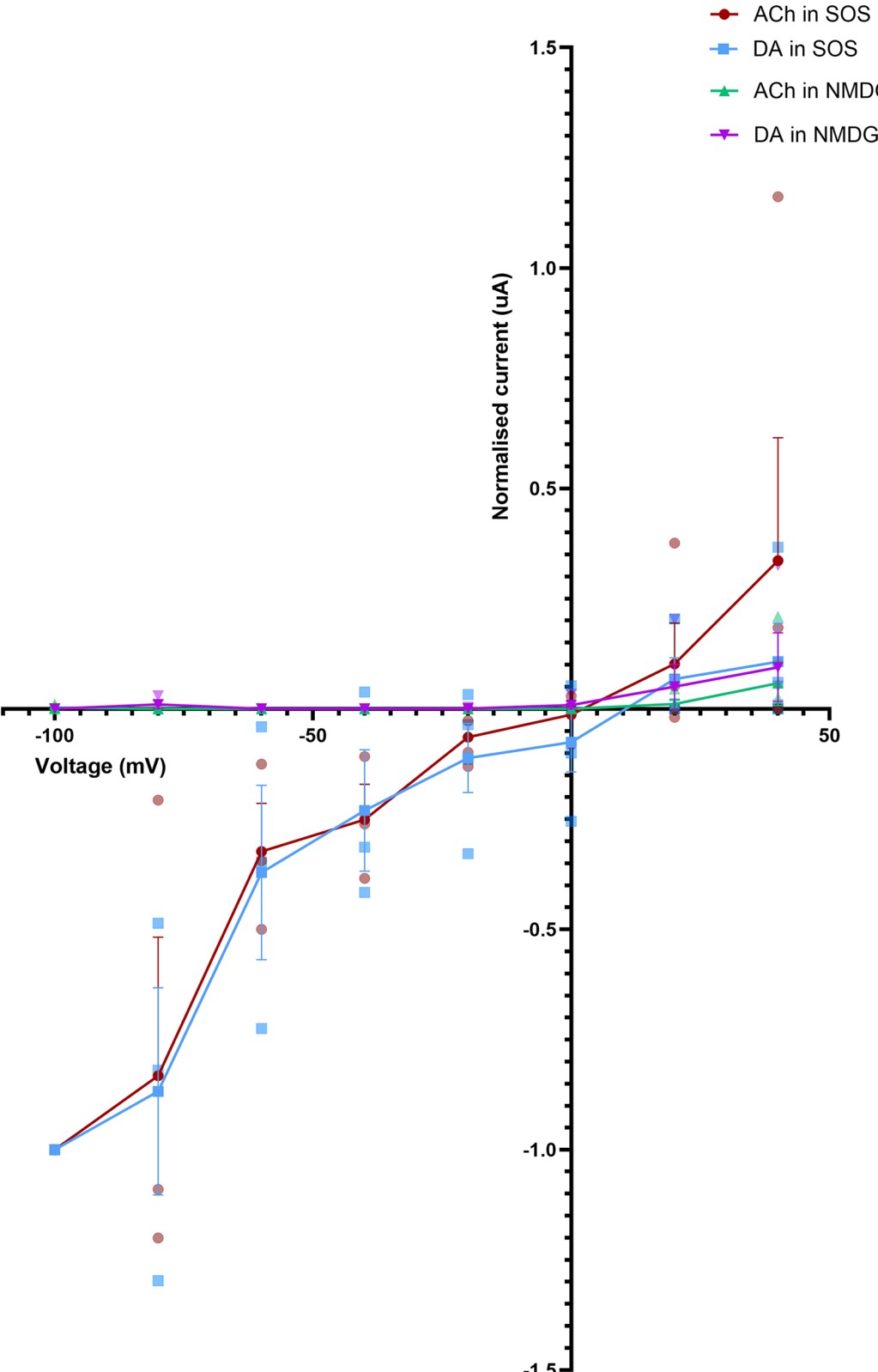

**Fig. 3 | The effect of a lack of sodium ions on the response of the *A. mellifera* a5 nAChR subunit expressed in *X. laevis* oocytes to agonists.** Current–voltage (I–V) plot for *A. mellifera* α5 injected oocytes responding to 2 mM acetylcholine (ACh) or 10 µM dopamine (DA) at voltages ranging between −100 mV and +40 mV perfused in either SOS buffer or NMDG solution, which lacks sodium ions. Values are normalised to the response to either 2 mM ACh or 10 µM DA at −100 mV. Data points are the mean of 3–4 different eggs from 2 batches of frogs. Error bars represent SEM.

**Fig. 4 | The effect of co-applying acetylcholine and dopamine on the *A. mellifera* α5 nAChR subunit expressed in *X. laevis* oocytes. a** Representative current traces showing the responses to DA (1 μM) and ACh (500 μM and 5 mM) independently, as well as co-applied. **b** Concentration curve induced by ACh (3 μM–10 mM) when co-applied with 1 μM DA. Values are normalised to 5 mM ACh + 1 μM DA. Data points are the mean from 3 to 5 different eggs from 2 batches of frogs. Error bars represent SEM. Scale bars represent time in seconds (X axis) and current μA (Y axis).

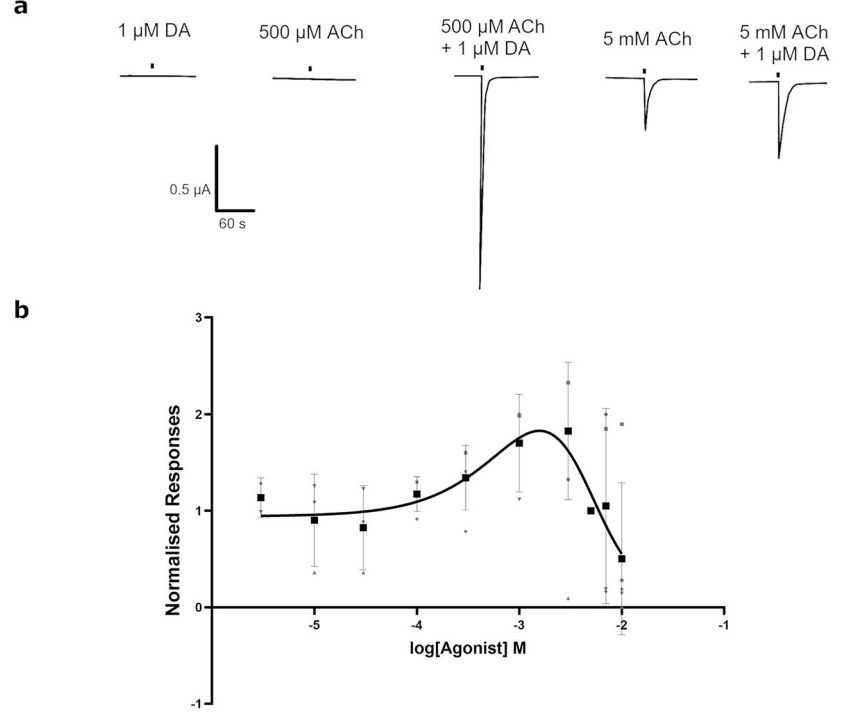

Diptera arose relatively late from gene duplication of the α7 nAChR subunit whilst the α5 subunit in non-Diptera was the result of a separate gene duplication event[19]. Attempts to express the *L. migratoria* α5 subunit as a homomeric receptor in *X. laevis* oocytes were unsuccessful but when co-injected with the rat β2 nAChR subunit, responses to ACh was observed[39]. This is in contrast to our findings that the *A. mellifera* α5 subunit readily expressed as a functional receptor when injected alone and did not seem to co-assemble with other *A. mellifera* subunits[19] indicating that α5 may require assembly with different subunits in a species-dependent manner.

The *A. mellifera* α5 subunit (originally referred to as Apis α7-2) was located by in situ hybridisation to the dorsal lobes, optic lobes, mushroom bodies (outer compact Kenyon cells) and antennal lobes[40]. This overlaps largely with the locations of serotonin, dopamine and octopamine found in the honey bee brain, although dopamine was not found in the Kenyon cells[41–43], indicating the possibility of the α5 nAChR reacting to biogenic amines in certain parts of the nervous system. Dopamine immunoreactive neurons were not found in the optic lobes however dopamine was isolated by HPLC from the optic lobes[41–43]. There has been less research on the location of tyramine as it was the last to be considered a neurotransmitter and the fact it is a substrate in the octopamine synthesis pathway makes it difficult to distinguish between octopaminergic and tyraminergic neurons[44,45]. However it has become clear that tyramine has distinct effects from octopamine, including in the honey bee with opposing actions in vision and there is a relatively high concentration of tyramine in insect brains[45,46].

It remains to be determined why the *A. mellifera* α5 receptor is particularly sensitive to dopamine instead of ACh. It may be that this receptor subtype mediates rapid response to biogenic amines present in nectar thereby modulating behavioural traits such as motivation, learning and reward-seeking that are important for plant-pollinator relationships[47]. It also remains to be determined whether other members of the non-Dipteran α5 nAChR subunit group are sensitive to biogenic amines, showing this group as a distinct subtype of receptors that have expanded from nAChRs to perhaps fulfil particular roles in certain insect species. Our study highlights how small changes in the complementary binding site[4] can have dramatic effects on receptor pharmacology with the potential to drive distinct avenues of cysLGIC evolution.

## Methods

### Reagents

Neomycin, amikacin, antibiotic antimycotic solution, collagenase type I from *Clostridium histolyticum*, ACh chloride, dopamine hydrochloride, tyramine, amitraz, γ-aminobutyric acid, glycine, homovanillyl alcohol and α-bungarotoxin were purchased from Sigma-Aldrich (Gillingham, UK). Calcium chloride and sodium chloride are from VWR Amresco Life Sciences (Lutterworth, UK) and potassium chloride from VWR Prolabo Chemicals (Lutterworth, UK). Magnesium chloride, histamine dihydrochloride, sodium L-glutamate monohydrate and octopamine hydrochloride were from Merck (Gillingham, UK). Dimethyl sulfoxide (DMSO) was from BDH Laboratory Supplies (Poole, UK). HEPES was from Melford (Ipswich, UK). NMDG was purchased from Thermo Fisher Scientific Inc. (Waltham, Massachusetts, U.S.)

### Animals

Oocytes were either obtained ready to inject from Ecocyte Europe (https://ecocyte-us.com/products/xenopus-oocyte-delivery-service/) or ovary tissue from adult female *X. laevis* was purchased from The European Xenopus Resource Centre based at Portsmouth University (Portsmouth, UK) or *X. laevis* frogs were purchased from Xenopus 1, Dexter, Michigan, USA. Frogs were handled strictly, adhering to the guidelines of the Scientific Procedure Act, 1986, of the United Kingdom.

### Expression of the *A. mellifera* α5 receptor in *X. laevis* oocytes and two-electrode voltage-clamp electrophysiology

Oocytes were mainly obtained either from Ecocyte (https://ecocyte-us.com/products/xenopus-oocyte-delivery-service/), which arrived ready for injection, or from the European Xenopus Resource Centre (https://xenopusresource.org/resources) as ovarian sections. In the latter case, oocytes were either prepared for injection by incubation in collagenase type I in OR2 (sodium chloride 82 mM, potassium chloride 2 mM, magnesium chloride 2 mM, HEPES 5 mM, pH 7.6) at 2 mg/ml for 45 min at room temperature and shaking at 150 RPM[19] or by manual defolliculation followed by incubation with 0.5 mg/ml collagenase type I in OR2 for 6 min at room temperature at 150 RPM. Oocyte nuclei were injected with 23 nl of *A. mellifera* α5 DNA cloned into the pCI vector[19] (Promega) at 300–500 ng/μl and stored in in standard oocyte saline solution (SOS; 100 mM sodium

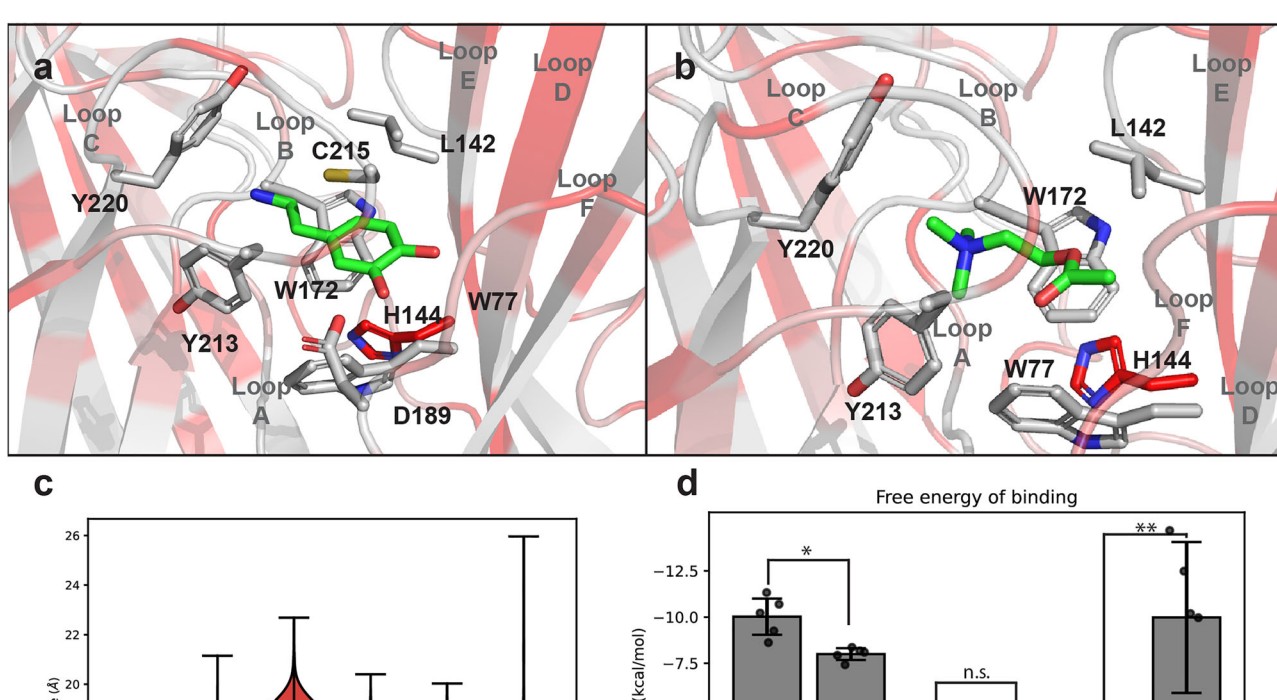

**Fig. 5 | Binding of dopamine (DA) and acetylcholine (ACh) to an *A. mellifera* α5 nAChR protein model.** Selected binding conformations of dopamine (**a**) and acetylcholine **b** rendered by PyMol[68]. The protein is coloured according to the conservation of the *A. mellifera* α5 nAChR sequence with the human α7 receptor. The colours are equivalent to sequence consensus symbols from light grey (full conservation '*') to red (full mismatch ' '). The ligand is shown in green. **c** Distributions of the distance between the centre of geometry of loop C and the centre of geometry of the protein for each ligand and structure. **d** Binding energies predicted by QM/MM GBSA. SH-PP denotes the S143P + H144P double mutant. $N = 5$ independent replicates. The *indicates a significant difference as determined by a two-tailed t-test with $p < 0.05$ and ** with $p < 0.01$. For ACh vs. DA $p = 0.011$, SH-PP ACh vs. SH-PP DA $p = 0.001$ and H144P ACh vs. H144P DA $p = 0.155$. Error bars represent SEM.

**Table 2 | Responses of *A. mellifera* α5 nAChR mutants expressed in *X. laevis* oocytes to acetylcholine (ACh) and dopamine (DA)**

| Mutant | ACh EC$_{50}$ μM | DA EC$_{50}$ μM | DA/ACh response |
|---|---|---|---|
| Wild-type | 2072 (773.6–5550) | 10.86 (4.409–26.73) | 14.94 ± 1.816 |
| E138H | 2760 (2386–3193) *$p = 0.0382$ | 53.05 (23.16–121.5) *$p = 0.0315$ | 9.370 ± 2.033 ****$p < 0.0001$ |
| V140Q | 1410 (672.2–2957) $p = 0.7279$ | 30.53 (23.24–40.10) **$p = 0.0067$ | 26.54 ± 5.504 **$p = 0.0034$ |
| S143P | NE | NE | NE |
| H144P | 434.7 (348.3–542.5) *$p = 0.0437$ | Too small responses | 0.504 ± 0.118 ****$P < 0.0001$ |
| S143P + H144P | NE | NE | NE |
| E138H + S143P + H144P | NE | NE | NE |
| E138H + V139C + V140Q +S143P + H144P | NE | NE | NE |
| human α7lpE | NE | NE | NE |
| human α7lpC | 2055 (1385–3049) $p > 0.9999$ | 35.37 (22.09–56.64) $p = 0.2925$ | 2.376 ± 0.343 **$p = 0.0040$ |
| human α7lpC + α7lpE | NE | NE | NE |

EC$_{50}$ values are displayed as the mean (95% confidence limits). DA/ACh response is the mean ± SEM ratio of the maximal responses to 100 μM dopamine and 5 mM acetylcholine. All data are from 5 to 6 different oocytes from ≥3 batches of eggs. NE indicates no expression was detected. 'Too small responses' indicates an EC$_{50}$ curve could not be generated. Statistical test used was one-way ANOVA with Bonferroni's multiple comparisons test.
*Indicates a significantly different value to that of wild-type ($p \leq 0.05$), **($p \leq 0.01$), ***($p \leq 0.001$) and ****($p \leq 0.0001$).

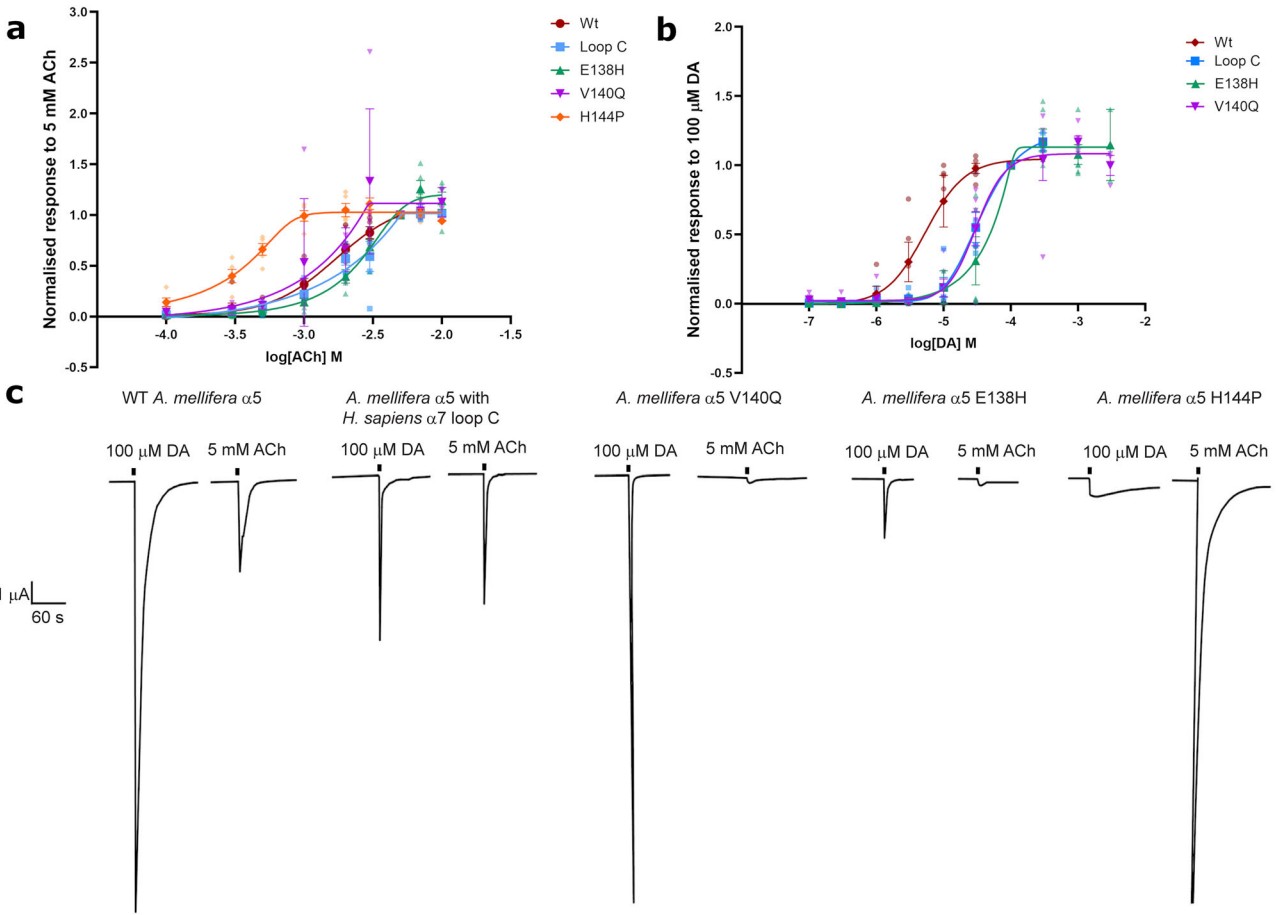

**Fig. 6 | Responses to dopamine (DA) and acetylcholine (ACh) in *X. laevis* oocytes expressing wild-type or mutant *A. mellifera* α5 nAChR subunits. a** Acetylcholine concentration response curves. Data are normalised to 5 mM acetylcholine. **b** dopamine response curves. Data are normalised to 100 μM dopamine. **c** Representative current traces showing responses to maximal concentrations, 100 μM dopamine and 5 mM acetylcholine, on the same oocytes. Concentration response curve data are from 5 to 6 different eggs from 3 to 5 batches of frogs. Error bars represent SEM. Scale bars represent time in seconds (X axis) and current μA (Y axis).

chloride, 2 mM potassium chloride, 1.8 mM calcium chloride, 1 mM magnesium chloride, 5 mM HEPES, pH 7.4) supplemented with 1× antibiotic antimycotic solution, 0.1 mg/ml amikacin and 0.05 mg/ml neomycin. Mutant *A. mellifera* α5 subunits (in pCI) were injected at 100–500 ng/μl. Oocytes were tested for responses 2–5 days after injection using two-electrode voltage clamp, with borosilicate glass microelectrodes filled with 3 mM potassium chloride (resistance 2–20 MΩ) and an Oocyte Clamp OC-725C amplifier (Warner Instruments, CT, USA). Oocytes were clamped at −80 mV and responses were recorded on a flatbed chart recorder (Kipp and Zonen BD-11E, Delft, The Netherlands). The oocytes were continuously perfused with SOS at a flow rate of 10 ml/min. Oocytes were selected for experiments if responses were consistent for two or more applications of the normalising concentration of agonist. Test chemicals were applied at 3 min intervals. Agonist concentration response curves were created by measuring the response of the oocytes to different agonist concentrations in SOS, responses were normalised to the maximal response induced by the agonist. Amitraz (10 mM) was dissolved in DMSO and then at a working concentration in 1:1000 ratio in SOS. All other chemicals were either dissolved in distilled water to make frozen stocks (100 mM) or dissolved directly in SOS (10 mM) on the day of use. To test receptor ion selectivity, NMDG solution (96 mM NMDG, 2 mM potassium chloride, 1.8 mM calcium chloride, 1 mM magnesium chloride, 5 mM HEPES, pH 7.4), which lacks sodium ions, was perfused for 3 min at a flow rate of 10 ml/min before applications of agonists diluted in NMDG solution. Oocytes were clamped at voltages ranging between −100 and +40 mV to determine the current voltage relationship.

Concentration-response data were fitted with the Hill equation:

$$y = I\max / \left\{ \frac{1}{\left(\frac{EC50}{[agonist]}\right)^{nH}} \right\}$$

where $y$ is the normalised current amplitude, Imax is the maximal response $\frac{I\max}{Iagonistmax}$, $EC_{50}$ is the agonist concentration at half-maximal efficacy, [agonist] is the agonist concentration and nH is the Hill coefficient, to estimate the $EC_{50}$ and Hill coefficient. Curve fitting was carried out using GraphPad software version 10 (GraphPad Software, La Jolla, CA, United States). All curves were plotted as log(agonist) vs response— variable slope after it became clear that curves with the Hill coefficient constrained to 1 did not give the best representation of the data. The peak of the current responses was normalised to the agonist concentration that gave a maximal response. Error bars on graphs show standard error, '*n*' indicates the number of experiments and '*batch*' indicates the number of frogs these oocytes came from. Values are shown as mean ± 95% confidence limits.

**Construction of *A. mellifera* α5 receptor mutants**

Point mutations in loop E of the *A. mellifera* α5 nAChR subunit were generated by site-directed mutagenesis using the QuikChange II site-directed mutagenesis kit (Agilent Technologies®) according to the manufacturer's protocol. The primers used are shown in Supplementary Table 1. The replacement of loop C of *A. mellifera* α5 with the equivalent region from

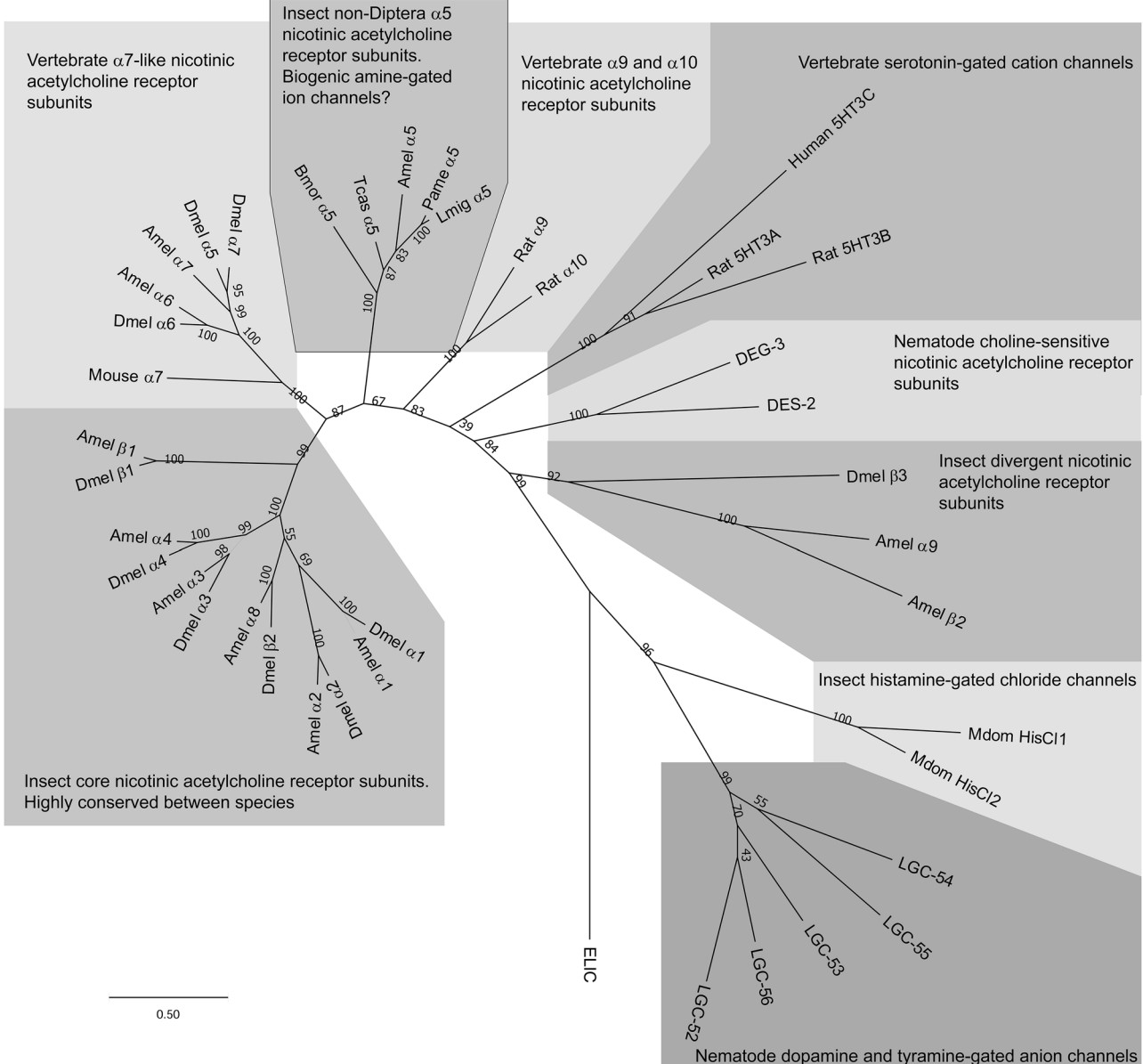

**Fig. 7 | Tree showing relationships of insect nicotinic acetylcholine receptor subunits with cys-loop ligand-gated ion channels activated by biogenic amines.** ELIC (Accession number P0C7B7), from Dickeya chrysanthemi, a bacterial ancestor of CysLGICs, was used as an outgroup. Peptide sequences were used to construct the phylogenetic tree with MEGA11 software[69] using the Maximum Likelihood method and Jones–Taylor–Thornton matrix model[70]. Numbers at each node signify boot-strapping 1000 times represented as a percentage of trees in which the associated taxa clustered together and the scale bar represents substitutions per site. Species subunit sequences used in the tree are as follows: *Apis mellifera* Amel α5 (AJE70263) otherwise see ref. 16; *Bombyx mori* Bmor α5 (EU082080); Drosophila melanogaster Dmel α1 (CAA30172), Dmel α2 (CAA36517), Dmel α3 (CAA75688), Dmel α4 (CAB77445), Dmel α5 (AAM13390), Dmel α6 (AAM13392), Dmel α7 (AAK67257), Dmel β1 (CAA27641), Dmel β2 (CAA39211), Dmel β3 (CAC48166); *Locusta migratoria* Lmig α5 (KF873584); *Periplaneta americana* Pame α5 (MW234341); *Tribolium castaneum* Tcas α5 (NM_001162522); *Musca domestica* Mdom HisCl1 (XP_005183540), Mdom HisCl2 (XP_005191543); *Homo sapiens* Human 5HT3C (NP_570126); *Mus musculus* Mouse α7 (AAF35885); *Rattus norvegicus* Rat α9 (NP_075219), Rat α10 (NP_072161), Rat 5HT3A (NP_077370), Rat 5HT3B (NP_071525); *Caenorhabditis elegans* DEG-3 (NP_001379138), DES-2 (AAC98095), LGC-52 (CAB60529), LGC-53 (NP_741945), LGC-54 (NP_001343648), LGC-55 (NP_507870), LGC-56 (NP_001382273).

the *Homo sapiens* α7 nAChR subunit was achieved by overlap extension PCR using the Q5® High-fidelity PCR Kit (New England Biolabs, Ipswich, MA, USA) and primers shown in Supplementary Table 2.

**Simulations using three-dimensional receptor models**
A previously generated *A. mellifera* α5 nicotinic ACh receptor homology model based on the human α7 nAChR was used[19]. To improve simula-tion performance only the extracellular portions of two adjacent subunits were extracted from the model (Supplementary Figs. 1 and 2). The ligands were modelled using Avogadro[48] and docked into the receptor with AutoDock Vina[49] to generate the starting ligand conformation. Protein parameters were generated with GROMACS 2021[50] using the Amber ff14SB force field[51]. Ligand parameters were assigned using AmberTools22[52] with GAFF force field[53] and then the system was sol-vated in a cubic box of TIP3P[54] and neutralised with 0.15 M NaCl. Then, an energy minimisation with the steepest descent algorithm was run for 5000 steps, followed by an NVT equilibration at 310 K and an NPT equilibration at 1 atm for 1000 ps each with the protein backbone and the ligand heavy atoms fixed by positional restraints. All simulations were run using GROMACS 2021 mdrun engine[50].

## Replica exchange with solute tempering (REST)

The REST method is an enhanced sampling technique in which the temperature of the solute portion of the system is scaled along a series of replicas to overcome energy barriers[55,56]. We used the replica exchange method implemented in PLUMED 2.8[57], which allows selection for the solute region in a flexible way. In this case, the solute region was defined as the ligand and 16 residues that line the binding site (Y115, S171, W172, T173, Y213, C215, C216 and Y220 of chain A and S58, W77, V78, T79, W141, L142, S143 and H144 of chain B). This allows for acceleration of the binding modes of the ligand without affecting the rest of the system. To avoid the ligand leaving the binding site at high temperature spherical flat-bottomed positional restraints were applied to the heavy atoms of the ligand that constrained its movements within a 10 Å radius from the starting conformation. A flexible implementation of the Hamiltonian replica exchange method, which allows selection of the 'heated' region in a flexible manner by scaling the force field parameters of the solute region, was used[58]. The reduced number of atoms selected for heating allows for enough overlap of the potential energy, which results in a good exchange probability. A table with the exchange probabilities for each system has been added to the Supplementary material (Supplementary Table 3). To make up for the absence of the other subunits, the carbon alpha atoms of five initial and final extracellular domain residues of each chain were held in place using harmonic restraints. This type of construct gives consistent results with a full extracellular domain containing all five subunits, with significant reduction of the computational cost[59]. The system was simulated for 200 ns in a range of temperatures from 310 to 1000 K, which were spread across 16 replicas and exchanges were attempted every 1000 steps. To analyse the ligand conformation, the trajectory at 310 K was clustered using the GROMOS algorithm[60], selecting the solute region heavy atoms for calculations. The pairwise root mean square deviation matrix between trajectory frames was calculated using MDAnalysis[61,62]. To analyse the closure of loop C, the distance between the centre of geometry of loop C alpha carbons (residues 211–220) and the centre of geometry of all alpha carbons in the structure was measured.

## QM/MM GBSA calculations

Using the selected snapshot from the REST simulations, five independent 20 ns unrestrained simulations were run and the final 10 ns were used for QM/MM GBSA analysis with a single trajectory approach. The GB-Neck2 model[63] with a salt concentration of 0.15 M and mbondi3 radii were employed for calculations. Additionally, the ligand and all residues within 5 Å from it were treated with quantum mechanics using the PM6-DH+ semi empirical Hamiltonian[64], which has been reported to be more efficient than other similar methods[65]. Two hundred frames were sampled from the analysed trajectory. These calculations were performed using the gmx_MMPBSA tool[66].

QM/MMGBSA calculations were performed in triplicate and the mean was plotted ± SD. Statistical comparison was performed using a pairwise t-test. Distance distribution overlap was calculated by estimating each distribution's probability density function and integrating over the product of each function combination tested. These calculations were carried out using the SciPy Python package (https://www.nature.com/articles/s41592-019-0686-2). To select representative features for the dynamics of the binding site, all alpha carbon distances between residues in loops A, B, C, D and F and residue 144 in loop E were measured. A representative distance for each loop was selected by calculating the information gain for each feature for classifying the trajectory into either wild-type or mutant (Supplementary Fig. 4). The mutual information score was calculated using the sci-kit learn package[67]. The simulations were extended to test the stability of the complex (Supplementary Fig. 5).

## Statistics and reproducibility

Comparisons of $EC_{50}$ and maximum response values were carried out using one way ANOVA with Bonferroni's Multiple Comparison test using GraphPad software version 10 (GraphPad Software, La Jolla, CA, United States). Sample sizes and replicates are given in figure legends where appropriate.

## Reporting summary

Further information on research design is available in the Nature Portfolio Reporting Summary linked to this article.

## Data availability

Modelling simulation files are deposited here: https://github.com/francoviscarra/bee_alpha5. Source data for graphs have been included in the Supplementary Data. Other datasets generated during and/or analysed during the current study are available from the corresponding author on reasonable request.

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

## Acknowledgements

This work was supported by funding from the Biotechnology and Biological Sciences Research Council (BBSRC) [grant number BB/M011224/1] and the Oxford Brookes University Nigel Groome studentship.

## Author contributions

E.L.M. performed electrophysiology experiments and contributed to writing of the manuscript; E.B.A performed site-directed mutagenesis and performed electrophysiology on the resulting mutants as well as contributed to writing of the manuscript; F.V. performed the molecular simulations and contributed to writing of the manuscript; I.B. provided guidance on the electrophysiology experiments and contributed to writing of the manuscript; P.C.B. provided guidance on the molecular simulations and contributed to writing of the manuscript; J.A.G. contributed to writing of the manuscript; A.K.J. conceived the idea for the project, performed the phylogenetic analysis and contributed to writing of the manuscript.

## Competing interests

The authors declare no competing interests.
