## [Transparent Peer Review File · Communications Biology]

Characterisation of an unusual nicotinic acetylcholine receptor subtype preferentially sensitive to biogenic amines

Corresponding Author: Dr Andrew Jones

This manuscript has been previously submitted at another journal. This document only contains information relating to versions considered at Communications Biology.

Version 0:

Reviewer comments:

Reviewer #1

(Remarks to the Author)

The paper titled "Characterisation of a novel nicotinic acetylcholine receptor subtype preferentially sensitive to biogenic amines" by Eleanor L. Mitchell et al. investigates the functional properties of homomeric alpha 5 nicotinic acetylcholine receptor from non-Dipteran invertebrate, *Apis mellifera*. The paper reports an unusual property of the receptor to have higher affinity towards dopamine, rather than acetylcholine.

Overall, the methodology and the results are convincing. The study contributes significantly to the field of cholinergic receptors. It also evokes a novel interest regarding sensitivities of other members of nicotinic acetylcholine receptors to alternative endogenous ligands.

The few comments regarding the paper are below:

1. Fig. 1 clearly suggests that water-injected oocytes produce dopamine-evoked current within uA range. It is clearly smaller than dopamine-evoked current in alpha5 -injected oocytes. However, it is still observed. It is worth including a paragraph into the discussion section to explain what endogenous receptors in oocytes could contribute to this current. Besides this, were other amines investigated in the paper (tyramine and octopamine) tested on water-injected oocytes? If so, were there any responses?

2. The authors investigated contribution of Loop E to the sensitivity to dopamine. They mutated non-conserved residues within bee alpha 5 subunit to the corresponding residues in human alpha7 subunit (E138H, V140Q, S143P, H144P). However, it looks like within loop E of alpha5 Val139 corresponds to cysteine in alpha 7 subunit. Did author consider this non-conserved site for their mutagenic studies? If so, what were the results? If not, were there any reasons to not consider this site? A paragraph touching on V139 would benefit the paper.

Reviewer #2

(Remarks to the Author)

In their manuscript "Characterisation of a novel nicotinic acetylcholine receptor subtype preferentially sensitive to biogenic amines" Mitchell et al. find a high sensitivity of the alpha5 nAChR subunit of the honey bee towards dopamine, tyramine and octopamine and also a higher efficacy of those amines in activating the receptor than acetylcholine has. Based on molecular dynamics simulations, they conclude that residues in a certain loop, E, are responsible for selectivity towards dopamine. This idea is further supported by mutation to proline residues in loop E, as found in other nAChR subunits, leading to loss of affinity for dopamine. Differences in efficacy, on the other hand, are discussed to be the result of different protein conformations, in particular a stabilisation of a conformation with capped loop C.

Not being an expert in that fields, I cannot comment on the quality of the experiments. I do, however, have concerns with the molecular dynamics simulations:

1000K is an extreme temperature. Aside from any realistic protein being fully denatured and all water being evaporated, the force field is not made for such extremes. Most likely only the constraints kept the protein intact.

Another problem with the temperature range used, is that with only 16 replicas the associated temperatures are so different that there will be hardly any overlap between replicas. How many exchanges did the authors observe? That is what is the acceptance ratio for exchanges?

The authors did not perform unbiased MD simulations at ambient temperature of the docked complexes, but instead went for replica exchange simulations in which the ligand was constrained right away. They have therefore not at all tested the stability of the complex, i.e. whether the ligands remain at the binding site found by docking, when not being constrained to do so. Such simulations must be added as a control. Prolonging the simulations used for subsequent QM/MM GBSA would likely do.

The model of the $\alpha 5$ nAChR subunit investigated in the present work is a homology model, based on human $\alpha 7$ nAChR. The fraction of conserved amino acid residues is not too great (please provide the percentage), in particular in the discussed loops C and E. To better visualise those differences, the authors should provide an overlap of the 3D structures of human and honey bee model as supplementary material.

Minor:

Some of the references in the Materials and method section are given by their doi. This is convenient, but a little confusing when all other references are given in the usual [1] style.

Reviewer #3

(Remarks to the Author)

In this manuscript, Mitchell et al. show that the nicotinic acetylcholine $\alpha 5$ subunit of the honey bee, *Apis mellifera* forms a novel homomeric receptor in *Xenopus laevis* oocytes that is gated by biogenic amines. Building on their previous work (PMID: 35249651), the authors present novel and surprising data that show that, while the *A. mellifera* $\alpha 5$ nAChR is activated by ACh, biogenic amines such as dopamine, tyramine, serotonin and octopamine are much more potent and efficacious agonists. The authors identified residues that may contribute to the unexpected agonist profile for the receptor based on molecular dynamics simulations. Using site-directed mutagenesis, they confirmed His144 on loop E as one of the critical residues underlying the heightened sensitivity of the receptor to dopamine. When His144 was mutated to proline, the corresponding highly conserved residue in human nAChR, the mutant receptor responded efficaciously to ACh, but not dopamine. As an ion channel electrophysiologist, I naturally find this work intriguing. However, considering that the authors have published some ground work on this receptor previously, I believe it is a missed opportunity to not provide a more comprehensive functional characterisation of this receptor in this current manuscript.

1. The receptor exhibits a mixture of conventional and unconventional features—inhibition by α -bungarotoxin, activation by ACh (albeit poorly), preference for biogenic amines, and insensitivity to insecticides (or in the case of imidacloprid and thiacloprid, weak inhibition in contrast to their typical agonist activity at other AMnAChRs). A few obvious questions that arise from these findings are:

- a. if the receptor is still cation selective like mammalian nAChRs, despite the well conserved TM2 sequences. The authors should consider performing experiments to determine ion selectivity by (1) substituting cations with NMDG and (2) substituting anions with methanesulfonate or gluconate in recording buffer.
- b. if other structurally related transmitters and amino acids such as melatonin, adrenaline, noradrenaline, tryptophan, phenylalanine and tyrosine can also activate the receptor. Given the lack of specificity of the receptor, have the authors considered the possibility that this is a chemosensory receptor that responds to a range of compounds found in food and other environmental/aversive stimuli (e.g., pheromones)?
- c. if these agonists work synergistically. Constructing the concentration-response curve of ACh in the presence of, e.g., 1 μ M dopamine can be helpful to determine how these ligands may regulate the function of this receptor when they are available to bind at the same time.
- d. if other classical nAChR blockers such as epibatidine, MLA and DHE also block the receptor. How about dopamine- and serotonin-related antagonists?

These are important questions to answer to help rationalise the physiological effects of receptor activation.

2. Based on the functional data of the His144Pro mutant, the authors claim that His144 is a molecular determinant of dopamine's potency and efficacy at AM $\alpha 5$ nAChR (line 184-185; page 8), even though they were not able to determine the EC_{50} for dopamine due to low current levels (Table 2). Based on the exemplar current traces presented in Figure 4C, however, I am not convinced that it is impossible to determine dopamine potency. 100 μ M dopamine elicited current amplitude around 1 μ A when applied on the His144Pro mutant, which is much larger than the current elicited by 5 mM ACh on the Glu138His and Val140Gln mutants. Yet, the potencies of ACh were determined for both mutants. The authors need to provide stronger data to support the claim that the His144Pro mutation does indeed negatively affect both potency and efficacy of dopamine. Along the same lines, many mutants generated appeared non-responsive. Have the authors tested dopamine or other biogenic amines at higher concentrations to be certain that there was no expression of the mutant

receptors?

3. Since there is evidence suggesting that the *A. mellifera* $\alpha 5$ nAChR subunit gene is expressed along with other nAChR subunit genes (PMID: 36478483), have the authors tried co-expressing the different subunits? This is an important experiment to help rationalise if the homomeric $\alpha 5$ nAChR or other heteromeric forms are the physiologically relevant receptor.

4. I see substantial room for improvement in the visuals of this manuscript.

a. Show continuous traces, if possible, to help readers gauge the quality of recordings.

b. Indicate duration of compound application clearly. The bars used currently are too thin to be seen, and bars are missing in Figure 4C.

c. The current traces look askew in general.

d. Figure 2A, middle panel, what are those spikes during compound application?

e. It is very challenging to distinguish the concentration-response curves. Please consider increasing the symbol sizes and/or use colours.

f. Figure 3A, please consider labelling Loop A-E, principal/complementary faces, indicate important residues such as His144 to help orientate readers.

g. Space can be used more efficiently. The current presentation of the traces occupies a considerable amount of vertical space. Please consider compressing them vertically and include relevant supplementary figures in the main figures. For instance, I can see Supplementary Figures 1 and 3 being incorporated into Figure 1/3/4. If space permits, it would be beneficial to include the structures of the ligands as well.

Version 1:

Reviewer comments:

Reviewer #1

(Remarks to the Author)

The authors addressed all comments sufficiently.

Reviewer #3

(Remarks to the Author)

The authors have addressed most of the comments in this revised manuscript. I only have a minor suggestion: the overarching goal of the study remains somewhat unclear. Is the primary aim to characterise the functional and pharmacological properties of this receptor with a goal to identify novel insecticides, or is it mainly to explore its evolutionary relationships with homologous receptors in other species? Clarifying this in the abstract and introduction would provide important context for the extensive data presented and better highlight the current knowledge gaps the study addresses.

Reviewer #1 (Remarks to the Author):

The paper titled "Characterisation of a novel nicotinic acetylcholine receptor subtype preferentially sensitive to biogenic amines" by Eleanor L. Mitchell et al. investigates the functional properties of homomeric alpha 5 nicotinic acetylcholine receptor from non-Dipteran invertebrate, *Apis mellifera*. The paper reports an unusual property of the receptor to have higher affinity towards dopamine, rather than acetylcholine.

Overall, the methodology and the results are convincing. The study contributes significantly to the field of cholinergic receptors. It also evokes a novel interest regarding sensitivities of other members of nicotinic acetylcholine receptors to alternative endogenous ligands.

The few comments regarding the paper are below:

1. Fig. 1 clearly suggests that water-injected oocytes produce dopamine-evoked current within uA range. It is clearly smaller than dopamine-evoked current in alpha5 -injected oocytes. However, it is still observed. It is worth including a paragraph into the discussion section to explain what endogenous receptors in oocytes could contribute to this current. Besides this, were other amines investigated in the paper (tyramine and octopamine) tested on water-injected oocytes? If so, were there any responses? **In Fig 1, water-injected produced no detectable response to dopamine as shown by a straight line. The figure has been clarified with a bar to show what oocytes are injected with water only and with**

alpha5. There were also no tyramine and octopamine responses when oocytes were injected with water alone.

2. The authors investigated contribution of Loop E to the sensitivity to dopamine. They mutated non-conserved residues within bee alpha 5 subunit to the corresponding residues in human alpha7 subunit (E138H, V140Q, S143P, H144P). However, it looks like within loop E of alpha5 Val139 corresponds to cysteine in alpha 7 subunit. Did author consider this non-conserved site for their mutagenic studies?

If so, what were the results? If not, were there any reasons to not consider this site? A paragraph touching on V139 would benefit the paper.

As the reviewer pointed out, we did not test V139C individually as this was not highlighted by our modelling, instead giving priority to E138H, V140Q, S143P, H144P. This decision is based on the observation that V139 has its side chain oriented away from the binding site, so we did not expect a change in this residue that would alter the interactions of the binding site with the ligand. As suggested, we have added some text touching on this in Lines 233-236.

Reviewer #2 (Remarks to the Author):

In their manuscript "Characterisation of a novel nicotinic acetylcholine receptor subtype preferentially sensitive to biogenic amines" Mitchell et al. find a high sensitivity of the alpha5 nAChR subunit of the honey bee towards dopamine, tyramine and octopamine and also a higher efficacy of those amines in activating the receptor than acetylcholine has. Based on molecular dynamics simulations, they conclude that residues in a certain loop, E, are responsible for selectivity towards dopamine. This idea is further supported by mutation to proline residues in loop E, as found in other nAChR subunits, leading to loss of affinity for dopamine. Differences in efficacy, on the other hand, are discussed to be the result of different protein conformations, in particular a stabilisation of a conformation with capped loop C.

Not being an expert in that field, I cannot comment on the quality of the experiments. I do, however, have concerns with the molecular dynamics simulations:

1000K is an extreme temperature. Aside from any realistic protein being fully denatured and all water being evaporated, the force field is not made for such extremes. Most likely only

the constraints kept the protein intact.

Another problem with the temperature range used, is that with only 16 replicas the associated temperatures are so different that there will be hardly any overlap between replicas. How many exchanges did the authors observe? That is what is the acceptance ratio for exchanges?

The reviewer correctly suggests that standard protein force fields are not made for sampling a thermodynamic ensemble at such extreme temperatures. In this case, we are using a flexible implementation of the Hamiltonian replica exchange method, which allows to select of the “heated” region in a flexible manner by scaling the force field parameters of the solute region(1). As detailed in the methods section, the solute region was defined as the ligand and 16 residues that line the binding site (Y115, S171, W172, T173, Y213, C215, C216 and Y220 of Chain A and S58, W77, V78, T79, W141, L142, S143 and H144 of Chain B). This allows for acceleration of the binding modes of the ligand without affecting the rest of the system. Moreover, the ligand was restrained to the binding site with a 10 Å flat-bottom restraint. This has been reported to reconstruct experimentally resolved structures from arbitrary docking poses (2). Additionally, similar protocols has been used for diverse purposes, from studying conformational landscapes of kinases (3), peptides (4), intrinsically disordered proteins (5) and RNA (6), to improving convergence of free energy calculations (7). Finally, the reduced number of atoms selected for heating allows enough overlap of the potential energy, which results in a good exchange probability. A table with the exchange probabilities for each system has been added to the supplementary information (Supplementary Table 3) and there is additional text in the Methods section (Lines 371-376).

The authors did not perform unbiased MD simulations at ambient temperature of the docked complexes, but instead went for replica exchange simulations in which the ligand was constrained right away. They have therefore not at all tested the stability of the complex, i.e. whether the ligands remain at the binding site found by docking, when not being constrained to do so. Such simulations must be added as a control. Prolonging the simulations used for subsequent QM/MM GBSA would likely do.

We have extended the simulations and added a figure of the RMSD to the Supplementary Material with a reference to this in the Methods section (Lines 406-407).

The model of the alpha5 nAChR subunit investigated in the present work is a homology model, based on human alpha7 nAChR. The fraction of conserved amino acid residues is not too great (please provide the percentage), in particular in the discussed loops C and E. To

better visualise those differences, the authors should provide an overlap of the 3D structures of human and honey bee model as supplementary material.

We have added the conserved amino acid percentages of loops C and E in the Results section (Line 152). We have included the suggested overlap figure in the Supplementary Material Fig. 2.

Minor:

Some of the references in the Materials and method section are given by their doi. This is convenient, but a little confusing when all other references are given in the usual [1] style. **These references have been changed to being numbered as with the other references (Lines 380 and 383).**

Reviewer #3 (Remarks to the Author):

In this manuscript, Mitchell et al. show that the nicotinic acetylcholine $\alpha 5$ subunit of the honey bee, *Apis mellifera* forms a novel homomeric receptor in *Xenopus laevis* oocytes that is gated by biogenic amines. Building on their previous work (PMID: 35249651), the authors present novel and surprising data that show that, while the *A. mellifera* $\alpha 5$ nAChR is activated by ACh, biogenic amines such as dopamine, tyramine, serotonin and octopamine are much more potent and efficacious agonists. The authors identified residues that may contribute to the unexpected agonist profile for the receptor based on molecular dynamics simulations. Using site-directed mutagenesis, they confirmed His144 on loop E as one of the critical residues underlying the heightened sensitivity of the receptor to dopamine. When His144 was mutated to proline, the corresponding highly conserved residue in human nAChR, the mutant receptor responded efficaciously to ACh, but not dopamine. As an ion channel electrophysiologist, I naturally find this work intriguing. However, considering that the authors have published some ground work on this receptor previously, I believe it is a missed opportunity to not provide a more comprehensive functional characterisation of this receptor in this current manuscript.

1. The receptor exhibits a mixture of conventional and unconventional features—inhibition by α -bungarotoxin, activation by ACh (albeit poorly), preference for biogenic amines, and insensitivity to insecticides (or in the case of imidacloprid and thiacloprid, weak inhibition in contrast to their typical agonist activity at other AMnAChRs). A few obvious questions that arise from these findings are:

a. if the receptor is still cation selective like mammalian nAChRs, despite the well conserved TM2 sequences. The authors should consider performing experiments to determine ion selectivity by (1) substituting cations with NMDG and (2) substituting anions with

*methanesulfonate or gluconate in recording buffer. **We have shown that it is cation selective using NMDG, a new figure showing this is included (Figure 3) accompanied by new text (Lines 120-126).***

*b. if other structurally related transmitters and amino acids such as melatonin, adrenaline, noradrenaline, tryptophan, phenylalanine and tyrosine can also activate the receptor. Given the lack of specificity of the receptor, have the authors considered the possibility that this is a chemosensory receptor that responds to a range of compounds found in food and other environmental/aversive stimuli (e.g., pheromones)? **We have tested homovanillyl alcohol, which has a structure similar to dopamine and is a component of the honey bee Queen mandibular pheromone. We found that the alpha5 receptor did not respond to this, as shown in the expanded Figure 2 (Fig. 2d). Text has been added to note this (Lines 102-104).***

c. if these agonists work synergistically. Constructing the concentration-response curve of ACh in the presence of, e.g., 1 μ M dopamine can be helpful to determine how these ligands may regulate the function of this receptor when they are available to bind at the same time.

We thank the reviewer for this suggestion. We found that co-applying acetylcholine and dopamine had a synergistic effect, as shown by an enhanced response when compared to acetylcholine or dopamine alone. We generated a concentration-response curve of varying acetylcholine concentrations co-applied with 1 μ M dopamine, which is shown in a new figure (Figure 4). Text has been added to note this (Lines 129-135).

*d. if other classical nAChR blockers such as epibatidine, MLA and DHE also block the receptor. How about dopamine- and serotonin-related antagonists? **There are certainly many more compounds we could test but we feel that the manuscript, with previous suggested additions, now presents a substantial characterisation of this receptor.***

These are important questions to answer to help rationalise the physiological effects of receptor activation.

2. Based on the functional data of the His144Pro mutant, the authors claim that His144 is a molecular determinant of dopamine's potency and efficacy at AM α 5 nAChR (line 184-185; page 8), even though they were not able to determine the EC₅₀ for dopamine due to low current levels (Table 2). Based on the exemplar current traces presented in Figure 4C, however, I am not convinced that it is impossible to determine dopamine potency. 100 μ M dopamine elicited current amplitude around 1 μ A when applied on the His144Pro mutant,

which is much larger than the current elicited by 5 mM ACh on the Glu138His and Val140Gln mutants. Yet, the potencies of ACh were determined for both mutants. The authors need to provide stronger data to support the claim that the His144Pro mutation does indeed negatively affect both potency and efficacy of dopamine. Along the same lines, many mutants generated appeared non-responsive. Have the authors tested dopamine or other biogenic amines at higher concentrations to be certain that there was no expression of the mutant receptors?

We have tried generating concentration-response curves for H144P but unfortunately the responses are too small and thus we were unable to determine the EC50. We have amended the Abstract and Discussion to say that this mutation affects efficacy but it is inconclusive whether it affects potency (Lines 24 and 223).

3. Since there is evidence suggesting that the *A. mellifera* $\alpha 5$ nAChR subunit gene is expressed along with other nAChR subunit genes (PMID: 36478483), have the authors tried co-expressing the different subunits? This is an important experiment to help rationalise if the homomeric $\alpha 5$ nAChR or other heteromeric forms are the physiologically relevant receptor. **We did try co-expressing with other subunits, which was previously published in Mitchell et al. 2022, we have noted this in the Introduction (Lines 62-64).**

4. I see substantial room for improvement in the visuals of this manuscript.

a. Show continuous traces, if possible, to help readers gauge the quality of recordings.

b. Indicate duration of compound application clearly. The bars used currently are too thin to be seen, and bars are missing in Figure 4C. **It is not possible to show continuous traces, but we have thickened the bars indicating agonist application in Figs 1, 2, 4 and now Fig. 6c (was Fig 4c).**

c. The current traces look askew in general. **We have straightened up the traces as best as we can (Figs. 1, 2, 4 and 6).**

d. Figure 2A, middle panel, what are those spikes during compound application? **We have removed these responses, which does not detract from our findings as we have kept the response to co-application with 400 μ M octopamine in Fig 2e that conveys the same finding. Removing these responses also allowed space to add responses of other agonists (see point g).**

e. It is very challenging to distinguish the concentration-response curves. Please consider increasing the symbol sizes and/or use colours. **We have lengthened the lines for the symbols to make them more easy to distinguish (Fig 1c).**

f. Figure 3A, please consider labelling Loop A-E, principal/complementary faces, indicate important residues such as His144 to help orientate readers. **We have implemented the changes to Figure 3A (now Figure 5a and 5b) labelling loops and important residues as suggested by the reviewer.**

g. Space can be used more efficiently. The current presentation of the traces occupies a considerable amount of vertical space. Please consider compressing them vertically and include relevant supplementary figures in the main figures. For instance, I can see Supplementary Figures 1 and 3 being incorporated into Figure 1/3/4. If space permits, it would be beneficial to include the structures of the ligands as well. **We have incorporated responses from Supplementary Figure 1 (to GABA, glycine and glutamate) into Figure 2 a-c and feel the spacing in the figures have improved as a result.**